# Comparative analysis of 1152 African-American and European-American men with prostate cancer identifies distinct genomic and immunological differences

Walter Rayford[1,17], Alp Tuna Beksac [2,17], Jordan Alger[3], Mohammed Alshalalfa [4], Mohsen Ahmed[2], Irtaza Khan[2], Ugo G. Falagario [2], Yang Liu[5], Elai Davicioni[5], Daniel E. Spratt[6], Edward M. Schaeffer [7], Felix Y. Feng [4], Brandon Mahal[8], Paul L. Nguyen[8], Robert B. Den[9], Mark D. Greenberger[1], Randy Bradley[10], Justin M. Watson[11], Matthew Beamer[3], Lambros Stamatakis[3], Darrell J. Carmen[12], Shivanshu Awasthi [13], Jonathan Hwang[3], Rachel Weil[2], Harri Merisaari[14], Nihal Mohamed[2], Leslie A. Deane[15], Dimple Chakravarty [2], Kamlesh K. Yadav[16], Kosj Yamoah [13], Sujit S. Nair [2] & Ashutosh K. Tewari [2✉]

Racial disparities in prostate cancer have not been well characterized on a genomic level. Here we show the results of a multi-institutional retrospective analysis of 1,152 patients (596 African-American men (AAM) and 556 European-American men (EAM)) who underwent radical prostatectomy. Comparative analyses between the race groups were conducted at the clinical, genomic, pathway, molecular subtype, and prognostic levels. The EAM group had increased *ERG* ($P < 0.001$) and *ETS* ($P = 0.02$) expression, decreased SPINK1 expression ($P < 0.001$), and basal-like ($P < 0.001$) molecular subtypes. After adjusting for confounders, the AAM group was associated with higher expression of *CRYBB2, GSTM3*, and inflammation genes (*IL33, IFNG, CCL4, CD3, ICOSLG*), and lower expression of mismatch repair genes (*MSH2, MSH6*) ($p < 0.001$ for all). At the pathway level, the AAM group had higher expression of genes sets related to the immune response, apoptosis, hypoxia, and reactive oxygen species. EAM group was associated with higher levels of fatty acid metabolism, DNA repair, and WNT/beta-catenin signaling. Based on cell lines data, AAM were predicted to have higher potential response to DNA damage. In conclusion, biological characteristics of prostate tumor were substantially different in AAM when compared to EAM.

[1] The Urology Group LLC, Memphis, TN, USA. [2] Department of Urology, Icahn School of Medicine at Mount Sinai, New York, NY, USA. [3] Department of Urology, Medstar Georgetown University Hospital, Washington, DC, USA. [4] Department of Radiation Oncology, University of California San Francisco, San Francisco, CA, USA. [5] Decipher Biosciences, San Diego, CA, USA. [6] Department of Radiation Oncology, University of Michigan, Ann Arbor, MI, USA. [7] Department of Urology, Northwestern University, Chicago, IL, USA. [8] Department of Radiation Oncology, Brigham and Women's Hospital, Boston, MA, USA. [9] Department of Radiation Oncology, Thomas Jefferson University, Philadelphia, PA, USA. [10] Haslam College of Business, University of Tennessee, Knoxville, TN, USA. [11] WellStar Urology, Marietta, GA, USA. [12] Georgia Urology, Atlanta, GA, USA. [13] Department of Radiation Oncology, Moffitt Cancer Center, Tampa, FL, USA. [14] Department of Clinical Medicine, University of Turku, Turku, Finland. [15] Department of Urology, Rush University Medical Center, Chicago, IL, USA. [16] Sema4, a Mount Sinai venture, Stamford, CT, USA. [17] These authors contributed equally: Walter Rayford, Alp Tuna Beksac. ✉email: ash.tewari@mountsinai.org

Prostate cancer (PCa) is the most common solid organ malignancy in men, with 174,650 new diagnoses and 31,620 deaths expected in 2019 in the United States alone. Significant racial disparities exist in PCa outcomes, with African-American men (AAM) experiencing a higher incidence (186.8 vs. 107.0 per 100,000) and mortality rate (40.8 vs. 18.2 per 100,000) than European-American men (EAM)[1].

Although previous studies have shown that multiple factors including, cultural, socioeconomic, psychosocial, and healthcare access disproportionately influence higher cancer burden and consequently poor disease outcomes in AAM[2,3], our knowledge of the extent to which tumor biology contributes to the reported level of disparities is limited. In addition, the lack of a systematic approach to characterize the inherent differences in tumor biology in AAM precludes the detection of molecular events that are enriched in this population[4–6].

Effective prognosis and personalized treatment regimens for PCa require identifying tumor-specific genomic factors and events and discovering disease-associated mechanisms. Although previous studies have demonstrated the existence of genomic differences between AAM and EAM men, the underlying mechanisms driving poor survival in AAM patients are not completely understood[7]. A number of mechanisms have been attributed to race disparities in PCa, including varying molecular subtypes, anatomic tumor location, dysregulation of oncogenic pathways, and the tumor microenvironment[4,8–10]. To unravel race-specific unique molecular pathways implicated in PCa, we leveraged a multi-institutional database and analyzed genomic differences between AAM and EAM.

## Results

**Baseline characteristics.** Clinico-pathologic characteristics of the study cohort are summarized in Table 1. Importantly, compared to EAM, AAM presented with significantly higher pretreatment prostate-specific antigen (PSA) levels (7.9 vs. 6.5 ng/ml, $P < 0.001$). AAM also had higher combined pathologic T3b and 4 stage (17.4% vs. 11.3%, $P = 0.001$) and had higher post-radical prostatectomy (RP) cancer of the prostate risk assessment postsurgical (CAPRA-S) scores (14.9% vs. 9.7%, $P < 0.001$). Finally, AAM also had higher genomic risk of metastasis compared to EAM (Decipher high-risk group 38.2 vs. 33.2%, $P = 0.04$).

**Prognostic implications of race.** To further study prognostic biomarkers in both races, we used 20 prognostic gene expression signatures previously reported to be associated with the worst outcome in PCa[11]. Overall, we established that AAM and EAM segregated well into prognostically variable groups (Fig. 1a). Furthermore, we found that in comparison across the two groups, Decipher and the average genomic-risk (AGR) score (mean of 19 signatures excluding Decipher) were strongly associated with grade groups (GGs) ($P < 0.001$) (Fig. 1b, c). GG4/5 AAM and EAM had higher Decipher scores compared to patients with GG1/2 and GG3. Interestingly, Decipher scores were higher in AAM only in the GG1/2 group ($P < 0.001$) but not in other groups (Fig. 1b). However, the AGR score was higher in EAM patients with GG4/5 ($P = 0.001$) but was not different in the remaining groups (Fig. 1c). These findings suggest that racial disparities are greatest in the very low and very high-GGs.

**Table 1 Baseline characteristics.**

| Variable | All | AAM | EAM | P value |
|---|---|---|---|---|
| Number of patients | 1152 | 596 | 556 | |
| Age at surgery, mean (IQR) | 63.8 (58, 68) | 63.3 (58, 67.7) | 64 (58, 68.7) | 0.083 |
| Preoperative PSA, median (IQR) | 7 (5, 11) | 7.9 (5.6, 12) | 6.5 (4.8, 10) | **<0.001** |
| RP grade group, n (%) | | | | |
| 1 | 53 (4.6) | 35 (5.9) | 18 (3.2) | **0.024** |
| 2 | 639 (55.5) | 333 (55.9) | 306 (55) | |
| 3 | 281 (24.4) | 131 (22) | 150 (27) | |
| 4 | 79 (6.9) | 49 (8.2) | 30 (5.4) | |
| 5 | 100 (8.7) | 48 (8.1) | 52 (9.4) | |
| pT stage, n (%) | | | | |
| T2 | 645 (56\) | 322 (54) | 323 (58) | **0.001** |
| T3a | 274 (23.7) | 130 (21.8) | 144 (26) | |
| T3b | 149 (12.9) | 88 (14.7) | 61 (11) | |
| T4 | 18 (1.5) | 16 (2.7) | 2 (0.3) | |
| pN stage, n (%) | | | | |
| N1 | 41 (3.6%) | 21 (3.5) | 20 (3.6) | >0.999 |
| N0 | 51 (4.4%) | 27 (4.5) | 24 (4.3) | |
| CAPRA-S risk[a] | | | | |
| Low | 208 (32.6) | 62 (22.4) | 146 (40.4) | **<0.001** |
| Intermediate | 287 (45) | 126 (45.5) | 161 (44.6) | |
| High | 143 (32.4) | 89 (32.1) | 54 (15) | |
| Decipher score, median (IQR) | 0.52 (0.36, 0.66) | 0.54 (0.38, 0.68) | 0.50 (0.34, 0.65) | 0.14 |
| Decipher group, n (%) | | | | |
| Low | 455 (39.5) | 215 (36) | 240 (43.2) | **0.04** |
| Average | 184 (15.9) | 153 (25.8) | 131 (23.5) | |
| High | 413 (35.8) | 228 (38.2) | 185 (33.2) | |

*AAM* African-American men, *EAM* European-American men, *CAPRA-S* cancer of the prostate risk assessment postsurgical, *IQR* interquartile range, *PSA* prostate-specific antigen, *RP* radical prostatectomy.
Bold values indicate statistical significance *p* < 0.05.
[a]Missing data.

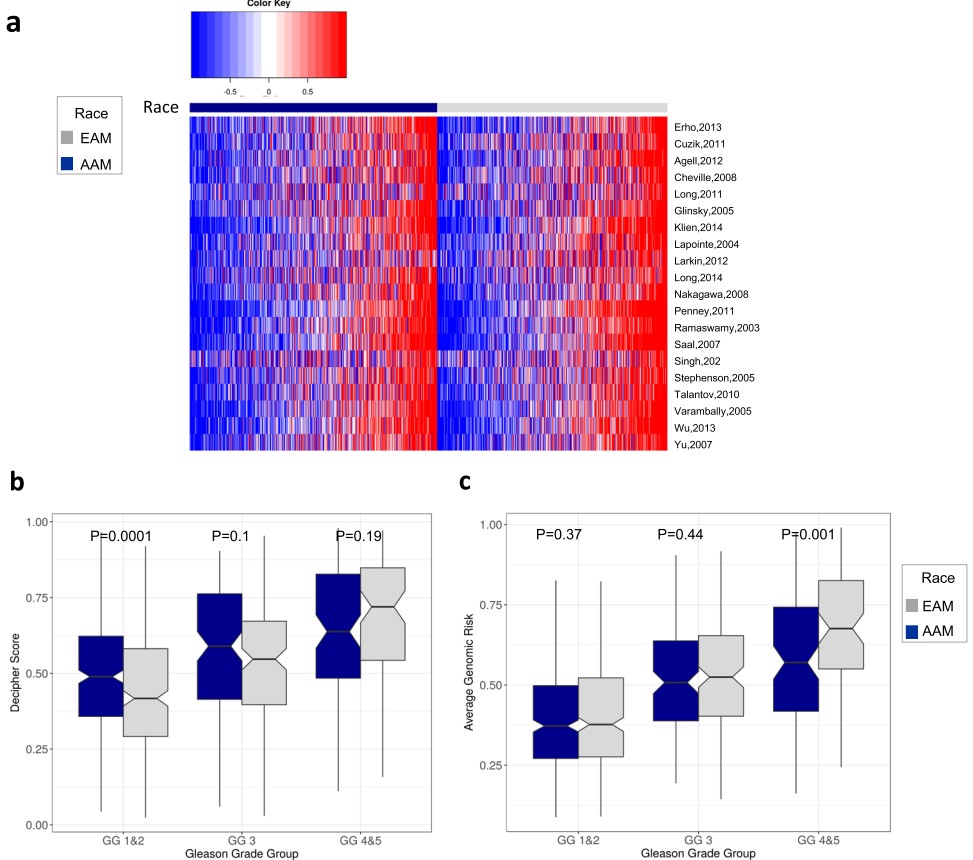

**Fig. 1 Comparison of genomic-risk signatures and their grade group associations in AAM and EAM. a** Prognostic differences in AAM vs. EAM. 20 prognostic signatures show similar trends in both groups. Overall, both Decipher (**b**) and average genomic-risk scores (**c**) positively correlate with grade groups in both races. **b** Decipher is higher in AAM with low grades but not different in high-grade tumors. **c** In contrast, average genomic risk is lower in the AAM with high-grade tumors. AAM African-American men, EAM European-American men. Error bars represent the 95% confidence interval.

**Molecular basis of racial disparities: genomic differences in EAM and AAM.** To understand the molecular underpinnings of racial disparities in PCa, we analyzed molecular subtypes of PCa within EAM and AAM groups based on the proportion of patients associated with ERG status and PAM50 subtypes, as previously defined[12]. Our findings demonstrate that AAM showed increased association with *SPINK1* expression (22% vs. 10%), EAM patients had higher ERG expression (41% vs. 23%) (Supplementary Fig. 1, top). In line with previous studies showing that the basal-like PAM50 subtype is associated with aggressive phenotype, we observed AAM tumors to be strongly associated with this subtype (42% vs. 35%) when compared to EAM tumors (Supplementary Fig. 1, bottom).

**Genome-wide differential expression between EAM and AAM.** Multivariable logistic regression was used to identify genes that were differentially expressed between the two groups after adjusting for pathological variables (GG, clinical T stage, lymph node involvement), and multiple testing. Using a false discovery rates (FDR) threshold of $1e^{-5}$, we found 4585 genes to be differentially expressed, with 778 overexpressed in the AAM group (Supplementary Fig. 2, Supplementary Data 2). The top differentially expressed genes in our cohort and TCGA are shown in Fig. 2a. Performing differential expression analysis using a Wilcoxon test in the TCGA-PCa cohort (37 AAM, 233 EAM), we found 35 genes (out of 560 genes upregulated in AAM in TCGA prostate) in common with the 778 upregulated genes in our cohort, and 295 genes (out of 646 genes with significant lower expression in AAM in TCGA prostate) in common with the 3807

genes downregulated in AAM in our cohort. We believe the small overlap between the two cohorts is first due to platform difference (RNA-seq vs. microarray) and difference in AAM sample size (37 in TCGA vs. 596 in our cohort).

*CRYBB2* is one of the most upregulated genes in the AAM group in both the TCGA prostate cohort and our cohort. Other genes, including *GSTM3*, *SNX31,* and *TENM1*, were among the top genes upregulated in AAM in both cohorts (Fig. 2b). *CRYBB2* and *PSPH* had been previously reported to be overexpressed in breast cancer samples[13] and colorectal cancer samples from AAM patients[14]. This suggests that these genes are overexpressed in AAM race regardless of cancer type or tissue of origin. A complete list of differentially expressed genes in both cohorts is provided in Supplementary Data 2.

**Signaling pathways driving the observed racial differences.** After characterizing genomic and gene expression differences, we next investigated the extent of differential regulation of key signaling pathways between the EAM and AAM groups. To achieve this goal, we first characterized the activity of cancer hallmark pathways in the AAM and EAM groups. Pathway activity was summarized as the mean expression of genes in the pathway. For every pathway score, we assessed its association with race using multivariable logistic regression models to calculated associated *P* values and odds ratios as described in the "Methods" section. Results from the multivariable models are shown in Supplementary Data 3. We observed mutual exclusivity in pathways and overall, AAM group tumors were associated with higher levels of immune cancer pathways, such as immune response

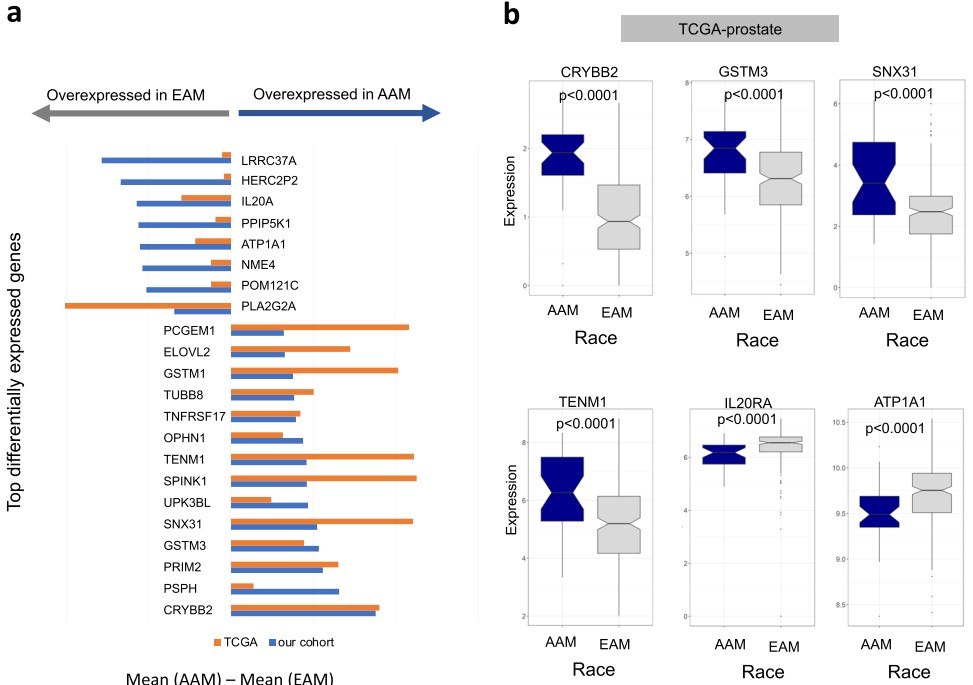

**Fig. 2 Genome-wide differential expression in AAM and EAM. a** Top 22 genes differentially expressed between the two groups in our cohort with multivariate analysis logistic regression *P* value <1e$^{-10}$. Mean difference in TCGA prostate was also shown for these genes to show that these genes show the same directionality. Genes were sorted based on the mean difference between AAM and EAM. **b** Boxplot of genes differentially expressed in TCGA prostate cohort. Abbreviations as in Fig. 1. Error bars represent the 95% confidence interval.

(tumor necrosis factor alpha signaling via NFκB, interferon alpha, gamma response), apoptosis, hypoxia, and reactive oxygen species, Fig. 3a (blue bars). In contrast, the EAM group was associated with higher gene pathway scores for MYC targets, DNA repair, and WNT-beta-catenin signaling (Fig. 3a, gray bars). Taken together, our data indicate that key biological processes associated with cell growth and survival pathways are upregulated in the EAM group. Our results define a role for immune signaling pathways in the AAM group and establish that upregulation of the immune-inflammation axis is a hallmark of AAM tumors.

**AAM have higher inflammation and immune responses**. To further characterize the biological impact associated with the upregulation of the immune signaling pathways in AAM, we conducted Gene Set Enrichment Analysis (GSEA) on genes upregulated in AAM using the EnrichR web-based tool. Enrichment signatures revealed significant association with the innate immune system, complement cascade and immunoregulatory interactions (*P* < 0.001 for all). To further characterize this association at the gene level, we studied the expression of several inflammation and immune checkpoint inhibitors genes. Interferon gamma (*IFN-G*), *CCL4*, and *IL33* were expressed at higher levels in the AAM group compared with the EAM group (*P* < 0.001) (Fig. 3b). The immune biomarker *CD3* was also expressed at significantly higher levels in the AAM group (*P* < 0.001), as were *ICOSLG* and *PDL2* (Fig. 3b). Our results are in line with a recent report analyzing pathway activity across race defined by genetic ancestry in TCGA prostate[5]. Using GSEA analysis in this report, the authors identified 18 gene sets were upregulated in AAM; 15 of 18 were immune-related signaling pathways. These findings support the hypothesis that AAM tumors are both more inflamed and more immune active.

**AAM have lower DNA repair**. Pathway analysis showed that AAM have lower DNA repair activity. We further investigated

other pathways and observed that the AAM group had significantly lower mismatch repair activity (Fig. 4a). To further identify DNA repair genes that are significantly driving this difference between the two groups, *MSH2*, *RAD52*, MSH6, and *PRKCD* were the most strongly downregulated DNA repair genes in the AAM group (Fig. 4b). *MSH2* and *MSH6* were also downregulated in AAM in TCGA prostate (*P* < 0.001 for both, Supplementary Fig. 3). Our results indicate that an aberrant DNA repair activity in AAM could have a significant impact on how AAM tumors respond to radiation therapy. To test this, we used a previously reported radiation response signature in PCa[15] and found that AAM indeed have higher signature scores (*P* < 0.0001) and thus are more likely to respond to radiation therapy. This observation needs to be tested in radiation therapy clinical trials.

**Validation of differential immune response, and immune response and AR response genes**. To further validate the differential expression of nominated targets in AAM vs. EAM on another platform, we used quantitative PCR to analyze the transcript profiles of representative genes that belong to DNA repair pathway (*MSH2*), inflammation (*CCL3*, *CCL4*, *IFNB*), and AR response genes (*RLN2, PCGEM1*). qPCR findings confirmed that genes associated with inflammation and AR response are upregulated in AAM and DNA mismatch repair gene, *MSH2* is downregulated in AAM (Supplementary Figs. 3 and 4).

**Transcriptomic heterogeneity of inflammation genes in AAM**. To further investigate the role of inflammation and immune genes in tumor progression in AAM, we characterized the expression of 124 tumor microenvironment and immune-response genes in the tumors from these patients. Consensus unsupervised clustering of AAM patients revealed a cluster of patients (22% of total AAM patients) who were enriched in adverse pathology features (defined by GG ≥3 and/or pN+ and/or pT3–4 disease; 60%) and high genomic risk (53%), and

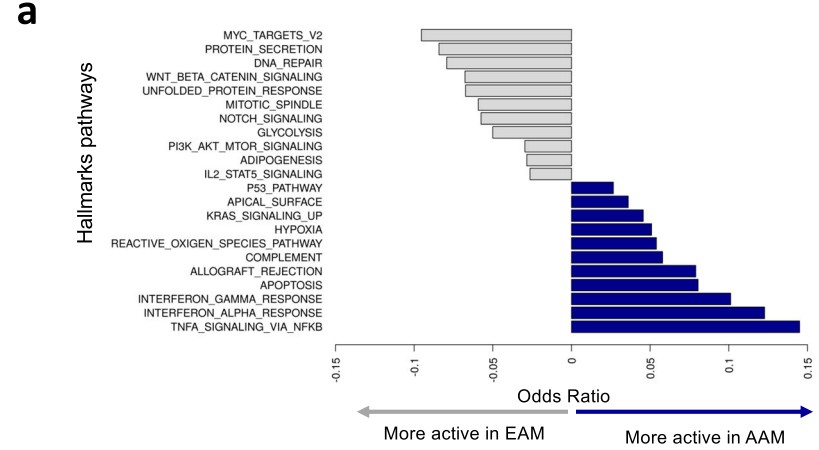

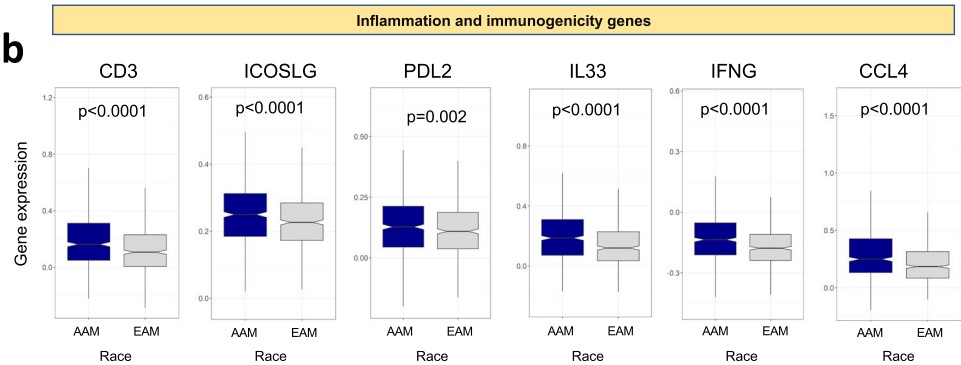

**Fig. 3 Activation of different signaling pathways drives prostate cancer in AAM and EAM. a** The bar plot depicts the log odds of pathways that have a significant association with AAM and EAM after adjusting for clinical variables and false discovery. Many of the pathways more active in the AAM are related to the immune response. Pathways more active in the EAM include those related to DNA repair, glycolytic metabolism, and the cell cycle. **b** African Americans showed higher inflammation activity and higher expression of *CD3, IFNG, IL33*, and immune checkpoint inhibitors (*PDL2, ICOSLG*). Abbreviations as in Fig. 1. Error bars represent the 95% confidence interval.

whose tumors had elevated expression of inflammation and immune-response genes, including *CD4, CD3, STAT1, GZMA, IDO1, CD2, CD96, PTPRC* (*CD45*), *CCL5, CXCL9, CXCL10*, and *CXCL11* (Fig. 5).

**Differential response to chemotherapy**. To test whether there was any segregation of chemotherapy sensitivity between the AAM and EAM, we evaluated predictions for response signatures of seven chemotherapy drugs that were previously developed using in vitro drug sensitivity and microarray data from the NCI-60 panel[16]. For each drug, we used the CellMiner tool[17] to explore NCI-60 data and identify drug response-related genes and their correlations to the IC50 value of NCI-60 cell lines. The most significantly correlated genes were selected and the expression of the corresponding genes in the Decipher Genomic Resource Information Database (GRID) were extracted for drug response score (DRS) calculations. A patient-specific DRS was calculated using these correlation coefficients (Cor) as weighting factors of the corresponding gene expression normalized by the sum of Cor. DRS was generated for 89 drugs across all 1152 tumor samples. These signatures predicted AAM men will have higher response score to DNA damage and alkylating agent-based chemotherapy and EAM men to potentially respond better to anti-microtubule-based chemotherapy (Supplementary Table 1). These drug response predictions based on cell lines need to be further investigated in clinical trials.

## Discussion

AAM men are underrepresented in PCa genomic profiling studies. For example, TCGA prostate have only 37 AAM (out of 333 patients). This small size of AAM samples has led to underpowered studies to identify the meaningful biological difference and is not enough to capture the heterogeneity in AAM populations. Thus, there is a need for cohorts enriched for AAM patients.

This study investigates biological differences between AAM and EAM is a large transcriptomic cohort with more than 50% being AAM. This study demonstrated that EAM and AAM groups have distinct genomic profiles, with clinical implications for managing active surveillance (AS), adjuvant treatment, recurrence management, and metastatic disease treatment. Characterizing the clinicopathological variables between AAM and EAM revealed that being AAM are associated with more aggressive clinical disease, including higher GGs, CAPRA-S, and Decipher scores. Observed distinctions in clinical and molecular characteristics within AAM and EAM can explain racial disparities in PCa. Our results provide compelling evidence suggesting that AAM considering RP should be carefully monitored post treatment.

We observed that AAM in our analytical cohort had a higher genomic risk of metastasis. Decipher and AGR scores correlated with GG of RP specimens. When stratified by GGs, AAM patients with GGs 1 and 2 had higher Decipher scores. In contrast, EAM patients with GGs 4 and 5 had higher AGR scores. These results could have clinical implications for AS management of AAM, as

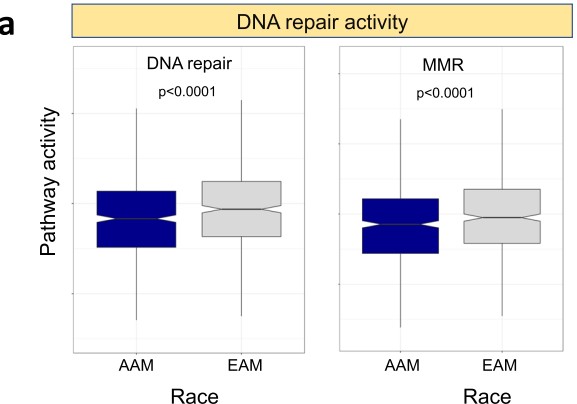

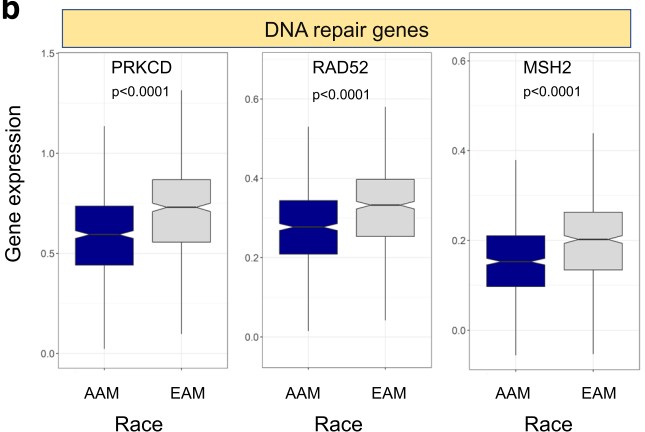

**Fig. 4 Differential expression of DNA repair pathways and genes. a** DNA repair activity and mismatch repair were calculated as mean of genes sets. Both showed lower activity in African Americans. **b** Key DNA repair genes (*MSH2, RAD52,* and *PRKCD*) were the most upregulated in patients. Error bars represent the 95% confidence interval.

AAM are twice as likely (HR, 2, 95% CI, 1.47–2.74, *P* < 0.0001) to be reclassified during AS[18] and have a higher genomic risk. Therefore, careful selection of AS is warranted for AAM men to avoid the risk of disease progression due to treatment delays. However, to make a more definitive recommendation, our results would need to be validated in an AS cohort. Furthermore, these results show the likely need for race-adjusted biomarker development.

An in-depth evaluation of molecular subtypes of PCa between EAM and AAM showed a significant association of *SPINK1* expression with AAM tumors. Our findings are consistent with previous studies validating that *SPINK1* was overexpressed among AAM[19]. In addition to *SPINK1*, we also observed significant differences in *ERG* expression by race. *AR*-mediated *TMPRSS2-ERG* fusion, an early event present in up to 70% of PCa, has been consistently shown to be absent in AAM[10,20]. Consistent with these findings, we observed that AAM patients had decreased *ERG* expression in our study. While the lack of *TMPRSS2-ERG* fusion in AAM has been validated, its prognostic value among AAM remains questionable[10,21,22]. Nonetheless, our results are consistent with published literature[21,23].

Furthermore, the PAM50 gene expression pattern demonstrated that AAM tumors are more enriched in basal subtypes. However, luminal B typing was similar between the two races. Because tumors with luminal B subtypes respond better to androgen deprivation treatment (ADT)[12], a similar expression profile between EAM and AAM would indicate similar responses to ADT responses between the groups.

Differences in tumor-associated intrinsic or extrinsic factors can potentially affect their response to drugs. Interestingly, analysis of drug response signatures showed that AAM men are predicted to have a higher potential response rate to alkylating agent-based chemotherapy, whereas EAM men have a higher potential response rate to taxane-based chemotherapy. These predictions require further validation in data from clinical trials. Taxane-based chemotherapy is the standard reference treatment in advanced disease[24], which, in theory, can result in favorable outcomes for EAM in the advanced disease cohort. Although this idea requires support from clinical outcomes data, it highlights the need to personalize treatment in patients with biochemical recurrence. Because alkylating agents work by causing DNA damage, this observation supports our earlier observation that AAM show increased susceptibility to DNA-damaging radiation therapy. If validated, these results could help personalize management of distant recurrence or metastatic disease in AAM. In contrast, taxane-based chemotherapeutic agents alter the microtubule network, thereby affecting chromosome segregation during cell division.

After adjusting for clinical variables and false discovery, we observed a differential activation of biological pathways in EAM versus AAM. EAM had higher rates of DNA repair, glycolytic metabolism, and cell cycle pathway activity, and we also observed a higher immune response. AAM also had higher inflammation activity, with higher T cell and IFN-γ responses and higher expression of *CD3E, CD3D, ICOSLG,* and *PDL2*. A recent report has shown that genes upregulated in AAM (ancestry based) in the TCGA prostate cohort are associated with immune-related signaling pathways[5]. This study supports our findings and further supports the rationale for using immunotherapy in AAM with PCa. Because novel PCa treatment options include immune modulation[23], understanding these unique racial differences is likely to understand the specific immune mechanisms better.

In fact, a subanalysis of the PROCEED trial showed a surprisingly high survival advantage for AAM receiving immunotherapy for metastatic castration-resistant PCa[25]. Our data suggest that AAM tumors have increased inflammation and immune-related functions and could be more responsive to immunotherapy regimens. Accordingly, we identified a subset of AAM tumors with molecular features that are likely to be more responsive to immunotherapy.

Our results could explain the underlying molecular mechanism of the differential response to radiotherapy and possible differences in outcome among AAM. Emerging studies have demonstrated that lower DNA repair is a prominent molecular feature of low AR activity which is linked with poor disease prognosis and response of radiotherapy[26,27]. Consequently, lower DNA repair activity can also underscore the molecular basis of higher radiation response among AAM with high-risk PCa[28]. Our findings are consistent with the meta-analysis of the RTOG trial that showed that AAM had better responses to radiation therapy due to differential AR signaling and DNA repair activity[26]. Although, we studied a RP cohort, we can extrapolate that gene signature and commercially available genomic test can individualize adjuvant treatment selection among AAM[29]. Further work in this regard is warranted.

The limitations of our study lie in its retrospective design and the inherent selection bias. Furthermore, our results are mostly descriptive, and we lack the data and sufficient follow-up to analyze oncological outcomes. The RNA expression data were derived from Decipher testing based on clinical indications; thus, our study represents a higher-risk cohort than the standard RP population. Therefore, our analytical cohort represents an ideal population in which to make assumptions regarding adjuvant or systemic treatment in the metastatic setting. Caution is

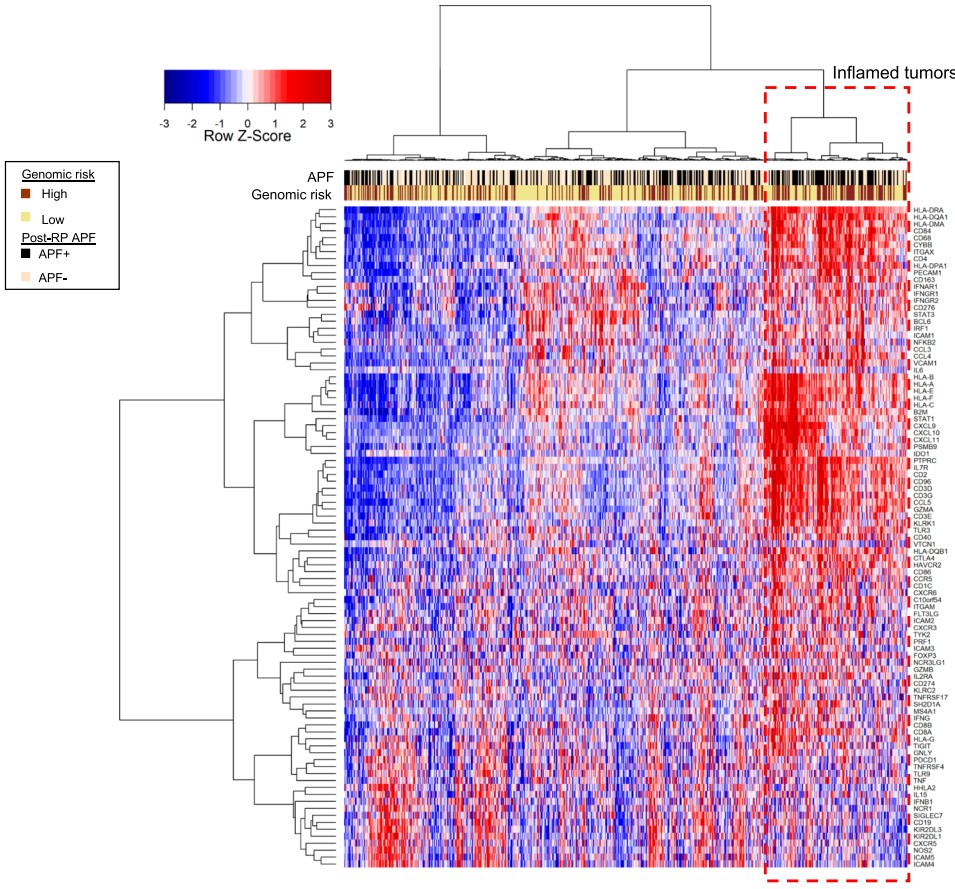

**Fig. 5 Tumor inflammation heterogeneity in AAM.** Using a set of 124 inflammation and immune-response genes in AAM, consensus clustering revealed a cluster of patients with high expression of immune genes (*CD3, CD45, GZMA, B2M, STAT1*), nominating this cluster for immunotherapy treatment intervention. AAM African-American men, APF adverse pathology feature.

recommended when applying clinical setting results to predict response to systemic therapy since patients were already treated with RP. Furthermore, unmeasured confounding resulting from socioeconomic factors can contribute to the baseline differences between the two races[30], and, therefore may impact the validity of our results. Regardless, this AAM-enriched cohort represents one of the largest multi-institutional analytical cohorts with complete molecular information to understand the distinct biologic features within race groups. In addition, a large sample size, along with extensive clinico-pathologic and genomic information, supports the strength and reproducibility of our results.

In conclusion, significant biological differences in PCa appear to depend, in part, on a man's racial ancestry, as well as, potentially his psychosocial and cultural environment. The inclusion of data from other racial groups will help further our understanding of the observed clinical heterogeneity of PCa and may have implications for personalized treatment in the future.

## Methods
**Data source and eligibility**. We performed an Institutional Review Board-approved (Icahn School of Medicine at Mount Sinai) retrospective analysis of de-identified genome-wide expression profiles in the GRID registry (NCT02609269) from clinical use of the Decipher Prostate RP test (Decipher Biosciences, San Diego, California). Expression data for each patient included the clinical Decipher score, as well as 20 pre-computed gene expression prognostic signatures and 56 hallmarks of cancer pathway activity scores, and genome-wide expression profiles using more than 46,000 gene and noncoding genes[31]. None of these patients received radiotherapy and/or androgen deprivation therapy before molecular testing. Overall, 1152 patients (596 AAM and 556 EAM) with available data on race and gene expression were included in this analysis.

**Clinical and pathology data**. Baseline demographic and clinical data included age, physician-reported patient race, and pretreatment PSA levels. Other pathologic variables, such as GG, pathological T (pT) stage and pathological N (pN) stage were also included. GGs were categorized based on the 2014 International Society of Urological Pathology Consensus Conference adopted GGs[32]. The CAPRA-S score was calculated as previously described[33]. Finally, patients genomic-risk classifier, Decipher score, was also used[34].

**Specimen collection and RNA expression profiling**. From RP specimens, formalin-fixed paraffin-embedded blocks were submitted for Decipher testing. Submitted tumor tissue included the highest GG with at least 0.5 mm² of tissue. RNA was extracted using the RNeasy kit (Qiagen, Santa Clara, California), and cDNA was prepared and amplified using the NuGEN Ovation WTA assay and hybridized to Human Exon 1.0 ST microarrays (Thermo-Fisher, Carlsbad, California), as described previously[34]. Microarray quality control was performed using Affymetrix Power Tools, as described previously[35]. Finally, probe-set summarization and normalization were performed using the single-channel array normalization algorithm[36], and the Decipher score was calculated as previously described[34].

For external validation of nominated targets in other platform qPCR, RNA isolated from fresh primary PCa tissues obtained from EAM and AAM (*n* = 11) who consented to PPHS/IRB study (Mount Sinai # GCO 14-0318) were used. Total RNA was prepared using Purelink RNA mini kit (Invitrogen) and reverse transcribed using the iscript Advanced cDNA synthesis reagent's following the manufacturer's instructions (BioRad Laboratories, USA). Tissues were homogenized in lysis buffer to ensure RNAse-free lysis and purified through a mini spin column. Following multiple washes, total RNA was collected into a final volume of 30 µl. Total RNA was measured using nano-drop and 260/230 and 260/280 ratios were considered as measure of quality. Quantitative real-time PCR was performed using SSO-Advanced Universal SYBR Green supermix (BioRad Laboratories, USA) and was analyzed on the CFX384 Touch real-time PCR system (BioRad Laboratories). Transcript levels were measured using the Delta-Ct method after normalization to seven housekeeping genes (*ACTB, HMBS, RPL38, TBP, GAPDH, HPRT, PSMC1*)[37,38].

**Pathway activity calculation and gene set enrichment analysis**. Pathway activity scores were calculated for hallmarks of cancer pathways (Supplementary Data 1) from the Molecular Signatures Database[39]. After scaling the expression of genes within each pathway gene set, the average of the scaled expression was used to represent pathway activity score as previously described[40]. An adjustment was made for baseline clinical differences between the two races using multivariable logistic regression. Finally, GSEA was conducted using the EnrichR online tool[41].

**Statistical analysis**. Analyses of the associations between categorical variables were conducted using the chi-square test or Fisher exact tests. The Mann–Whitney $U$ test was used as to evaluate the difference between non-normally distributed continuous variables. Multivariable logistic regression analysis adjusting for GG, pT stage and lymph node involvement was used to predict race (binary response variable) from hallmarks pathway activity scores (continuous predictor variable). Logistic regression here was applied for every pathway/gene individually to find a statistical association between race and pathway activity or gene expression using the following formula:

"$\log(p/1 - p)$ ($p = 1$: AAM, 0: EAM) $= \beta0 + \beta1 \times \text{pathway} + \beta2 \times \text{Gleason} + \beta3 \times \text{EPE} + \beta4 \times \text{SVI} + \beta5 \times \text{LNI} + \varepsilon$", where $\log(p/1 - p)$ is a logit function and $\varepsilon$ is an error for logistic regression formula. Then, the $P$ value and odds ratio of $\beta1$ was evaluated for all pathways. $P$ values of multiple comparisons across pathways were controlled for FDR using the Benjamini–Hochberg procedure.

**Reporting summary**. Further information on research design is available in the Nature Research Reporting Summary linked to this article.

## Data availability

Gene expression data used in this study are available upon request from the PI of this study (A.K.T.). Gene expression data are also freely available from the Decipher GRID upon request from Decipher Biosciences Inc. The microarray data that support the findings of this study were deposited on NCBI Gene expression Omnibus (GEO) and are accessible through GEO accession number GSE169038. Source data for Fig. 2 are available in Supplementary Data 4. A subanalysis of our work is in part based upon data generated by the TCGA Research Network: https://www.cancer.gov/tcga.

## Code availability

R Codes used in this study are created by M.A. and codes are available to be shared based on request. Metadata and R code used to generate the figures is available in Supplementary Data 5.

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

## Acknowledgements

The authors wish to thank Ms. Sima Rabinowitz for editing the manuscript. The authors wish to acknowledge funding support from the Deane Prostate Health, ISMMS and the Arthur M. Blank Family Foundation to A.K.T.

## Author contributions

Conception and design: W.R., A.T.B., M.A., E.D., K.K.Y., S.S.N., and A.K.T. Development of methodology: A.T.B., M.A., and E.D. D.C., K.K.Y., S.S.N., A.K.T. Analysis and interpretation of data; writing, review, and/or revision of the manuscript: W.R., A.T.B., J.A., M.A., M.A., I.K., U.G.F., Y.L., E.D., D.E.S., E.M.S., F.F.Y.F., B.M., P.L.N., R.B.D., M.D.G., R.B., J.M.W., M.B., L.S., D.J.C., S.A., J.H., R.W., H.M., N.M., L.A.D., D.C., K.K.Y., K.Y., S.S.N., and A.K.T. Feedback: All authors reviewed the manuscript, provided critical feedback and approved the manuscript.

## Competing interests

The authors declare the following competing interests: Y.L. and E.D. are employees of Decipher Biosciences. R.B.D. has received research funding and has served as a consultant for Decipher Biosciences. P.L.N. has served as a consultant for Decipher Biosciences, Ferring, Bayer, Astellas Medivation, Dendreon, Blue Earth, Nanobiotix, Augmenix, and Infinity Pharmaceuticals. F.F.Y.F. is an employee of PFS genomics and has served as a consultant for Medivation/Astellas, Decipher, Celgene, Dendreon, EMD Serono, Janssen Oncology, Ferring, and Bayer. D.E.S. has served as a consultant for Dendreon. E.M.S. has served as a consultant for Decipher, OPKO Health, Abbvie. K.K.Y. is an employee of Sema4, a Mount Sinai venture company, and has received royalties from Inthera Bioscience. A.K.T. has served as a site-PI on pharma/industry-sponsored clinical trials from Kite Pharma, Lumicell Inc, Dendreon, and Oncovir Inc. He has received research funding (grants) to his institution from DOD, NIH, Axogen, Intuitive surgical, AMBFF, and other philanthropy. A.K.T. has served as an unpaid consultant to Roivant Biosciences and advisor to Promaxo. He owns equity in Promaxo.
