## [Peer Review File · Communications Biology]

Reviewers' comments:

Reviewer #1 (Remarks to the Author):

The authors performed a multi-institutional retrospective analysis of 1,152 patients (596 African-American and 556 European-American men) who underwent radical prostatectomy. They found that African-American (AA) group had a significantly higher median pre-operative prostate-specific antigen. The European-American (EA) group had increased expression of ERG and ETS, whereas the AA group had more triple-negative, SPINK1, and basal-like molecular subtypes. Additionally, the AA group was associated with higher expression of the oncogenes (FOS and JUN) and inflammation genes (IL33, IFNG, CCL4, CD3, and ICOSLG). At the pathway level, the AA group had higher expression of genes sets related to the immune response, apoptosis, hypoxia, and reactive oxygen species. However, EA group tumors were associated with higher levels of fatty acid metabolism, DNA repair, and WNT/beta21 catenin signaling. The AA group had lower mismatch and DNA repair activity and lower expression of the PRKCD and MSH2 genes but had higher expression of inflammation and immune-response genes. Characterizing the clinicopathological variables between AA and EA patients may help us to improve RP post-treatment as well as personalize treatment, especially in AA patients. However, specific concerns about the manuscript are as below:

Major concerns:

- All data were generated by the Affymetrix Microarray. Also, there is no information on how to obtain the RNA samples from patients for molecular testing and how about the RNA quality of samples from a multi-institutional collection. Although some genes were validated in qPCR Analysis, most of the identified genes appear not to be supported by the analysis in TCGA datasets.
- Most of the patients are likely in early-stage as shown in Table 1. The tumor is heterogeneity, including both tumor cells and tumor microenvironment (TME) cells. For example, the extracted RNA derived from both tumor cells and TME cells. It is unclear whether the data is for both tumor cells and TME cells.

Minor concerns:

- The TCGA dataset possesses a large sample size, comprehensive molecular profiles, and clinical outcome information. An analysis with the TCGA data would be interesting to judge the results in the present study.
- There should be a clear delineation of what is the discovery in the text.
- The authors do not sufficiently discuss/reconcile conflicting observations with other reports.
- It would be informative to have IHC staining of the identified key genes in some cases, which will validate the relationship as well as racial disparities in prostate cancer.
- The cohort description is insufficiently relative to information on sample collection. Please detail how many patients were enrolled from each institute.

Reviewer #2 (Remarks to the Author):

The authors present a comparative study between prostate cancer patients of European and African ancestry, using expression data generated by the commercial Decipher test. The subject of health disparity is important and any additional information on this topic might be relevant.

Although the authors performed a number of analyses, these results are descriptive and not very conclusive. Some additional datasets might be tested for comparison, such as TCGA.

The validation set is not described at all (numbers, clinical and demographic characteristics, frozen/FFPE samples, RNA quality, experimental details, etc)

Methods should present more information, including on all the metrics used in the analyses. It should not take going through references to understand different scores and how they were generated, such as Decipher score, 82 pre-computed gene expression signatures, CARPA-score,

pathway activity scores, 20 prognostic signatures, etc. The NCI-60 gene signature analysis is not described in methods at all.

Supplementary figure 3 should explain error bars and provide p-values

Reviewer #3 (Remarks to the Author):

In this manuscript the authors carried out retrospective study by calculating pathway activity scores and multivariable logistic regression analysis using gene expression data derived from microarrays (>46,000 genes), gene expression signatures (82 genes) and Decipher score of 1,152 African-American (AA) and European-American (EA) with prostate cancer who underwent radical prostatectomy. The authors presented results that showed a higher median pre-operative PSA, more triple negative (?), SPINK1 and basal like molecular subtypes among AAs and increased ERG expression among EA. Prostate tumors of AA men were also associated with higher expression FOS and JUN oncogenes, and genes related to inflammation, immune response, hypoxia and reactive oxygen species but lower expression of DNA damage repair and DNA mismatch repair genes. Tumors of EA men, in contrast, were associated with higher levels of expression for fatty acid metabolism, DNA repair and beta catenin signaling. The authors conclude that significant biological differences in prostate cancer appear to be dependent on a man's racial background.

This manuscript proposed that adjuvant treatment after radical prostatectomy can be extrapolated from gene expression signature of prostate tumors. To achieve this the authors described a series of data analyses that associate gene expression profiles, cell signaling pathways, and in vitro drug sensitivity and microarray data from the NCI-60 panel. Overall this idea is original. The manuscript, however, failed to communicate this convincingly because it was neither emphasized in the abstract nor explained clearly in the introduction. Furthermore, missing incomplete citations flagged by placeholders suggest poor editing. The manuscript could be improved if the concerns listed below are addressed .

Contrary to the authors assertion (abstract, line 2-3), several studies have characterized racial differences in post prostate cancer at the genomic level through whole exome sequencing and whole genome sequencing of prostate tumors from African American patients. The authors should cite these reports and include them in the introduction or discussion.

Additional primary references and a link to the website describing the genomic resource information database (GRID) (line 46-53) should be cited.

The authors should provide the number of European American and African American subjects used in the validation assay described on Page 4 (line 72-73), and describe which housekeeping genes used for normalization of PCR assays (Page 6, line 77).

Table 1 describing the baseline characteristics should include bladder neck invasion and seminal vesicles invasion (Line91-95).

Line 96-97/Figure 1: the authors should describe the criteria for selecting the 20 prognostic signatures used in Figure 1 and refer to these reports in the main text.

It would be helpful if the authors should clarify what does the "triple" in triple negative subtype refer to in Supplementary Figure 1 (line 106-114) and in the abstract.

The authors should include corrected p-values for multiple hypothesis testing to account for false positives or type I error of the multivariate logistic regression analysis in Figure 2 and tabulate the results in the Supplementary data (Page 6, Line 114-124)

It would be helpful if the source or specific gene ontology library used in association with the regression coefficients analysis described on page 7 is mentioned and representative genes that

define cell growth and survival pathways, as well as the immune inflammation axis could be listed (Line 126-127).

It would be helpful if the 82 upregulated genes identified by multivariate logistic regression, together with the p-values correcting for multiple hypothesis testing to account for false positives or type one error were presented in the supplementary data (Line 138-144) Supplementary table 1 was not mentioned in the text.

In addition to reference 27, it would be more relevant if the authors could cite similar epigenetic studies involving human subjects.

The authors should cite relevant references on the factors that potentially contribute to racial health disparities listed in 250 to 255.

The authors should replace the placeholders for references cited at the end of sentences in line 259-262 and 270 with the actual references

Reviewers' comments:

Reviewer #1 (Remarks to the Author):

The authors performed a multi-institutional retrospective analysis of 1,152 patients (596 African-American and 556 European-American men) who underwent radical prostatectomy. They found that African-American (AA) group had a significantly higher median pre-operative prostate-specific antigen. The European-American (EA) group had increased expression of ERG and ETS, whereas the AA group had more triple-negative, SPINK1, and basal-like molecular subtypes. Additionally, the AA group was associated with higher expression of the oncogenes (FOS and JUN) and inflammation genes (IL33, IFNG, CCL4, CD3, and ICOSLG). At the pathway level, the AA group had higher expression of genes sets related to the immune response, apoptosis, hypoxia, and reactive oxygen species. However, EA group tumors were associated with higher levels of fatty acid metabolism, DNA repair, and WNT/beta21 catenin signaling. The AA group had lower mismatch and DNA repair activity and lower expression of the PRKCD and MSH2 genes but had higher expression of inflammation and immune-response genes. Characterizing the clinicopathological variables between AA and EA patients may help us to improve RP post-treatment as well as personalize treatment, especially in AA patients. However, specific concerns about the manuscript are as below:

Major concerns:

- All data were generated by the Affymetrix Microarray. Also, there is no information on how to obtain the RNA samples from patients for molecular testing and how about the RNA quality of samples from a multi-institutional collection. Although some genes were validated in qPCR Analysis, most of the identified genes appear not to be supported by the analysis in TCGA datasets.

All these samples were from the clinical routine use of the Decipher test. Central pathology review was performed for all samples. All samples specimen selection, RNA extraction, and microarray hybridization were done in a Clinical Laboratory Improvement Amendments (CLIA)-certified laboratory facility (GenomeDx Biosciences, San Diego, CA, USA). Thus, having these samples from multi-institutional cohort doesn't affect the analysis of the data as all samples were processed at GenomeDx facility from samples selection to data generation.

We have now conducted differential expression analysis between African American and Caucasians in the TCGA-prostate cohort and found that top upregulated genes in our cohort from Figure 2 (CRYBB2, PRIM2, GSTM3, SNX31, TENM1, GSTM1) were also significantly upregulated in African American in TCGA-prostate. Similar result was observed in the downregulated genes. Out of the 646 genes downregulated in African American in TCGA-prostate, 295 (45%) genes were also downregulated in our cohort. We believe that this moderate overlap is due to the proportions of African American in the cohort. TCGA has ~10% African American, while our cohort has 50%. Thank you for your comment.

- Most of the patients are likely in early-stage as shown in Table 1. The tumor is heterogeneous, including both tumor cells and tumor microenvironment (TME) cells. For example, the extracted RNA derived from both tumor cells and TME cells. It is unclear whether the data is for both tumor cells and TME cells.

Response: We thank the reviewer for this point. Although we agree that our cohort is mostly consisting of local or locally advanced patients, we do have a high proportion of high risk patients (PSA>20 or GS>7 or pT3 or 4 or SM+), whose clinical scenarios warranted a genomic risk evaluation. With regard to tumor sampling, the acceptance criteria for the Decipher assay include at least 0.5 cm² of tumor with at least 60% neoplastic cells. This is no difference than TCGA cohorts which use threshold of tumor purity of 60%.

Minor concerns:

- The TCGA dataset possesses a large sample size, comprehensive molecular profiles, and clinical outcome information. An analysis with the TCGA data would be interesting to judge the results in the present study.

We have now conducted differential expression analysis between African American and Caucasians in the TCGA-prostate cohort and found moderate overlap between upregulated and downregulated genes in both cohorts, but the top genes in our cohort were also significantly different in TCGA.

Even though TCGA-prostate has comprehensive molecular profile, there is only ~40 (10%) African American patients, while our cohort has more than 500 African American patients. So, we think our cohort is way more powerful to conduct transcriptome-wide analysis. Thank you for your contribution.

- There should be a clear delineation of what is the discovery in the text.

We now edited the manuscript to highlight the take-home message from this paper

- The authors do not sufficiently discuss/reconcile conflicting observations with other reports.

We now discussed more our results and how they explain observations in recent clinical trials

- It would be informative to have IHC staining of the identified key genes in some cases, which will validate the relationship as well as racial disparities in prostate cancer.

We thank the reviewer for this point, although we agree that IHC staining could give additional information, IHC staining is beyond the scope of this paper.

- The cohort description is insufficiently relative to information on sample collection. Please detail how many patients were enrolled from each institute.

We now added more details on the source of the samples. Approximately 50% came from The Mount Sinai Hospital, 27% from The Urology Group practice in Memphis, and 15% from Medstar Georgetown University Hospital. Remaining patients came from

smaller practices throughout USA.

Reviewer #2 (Remarks to the Author):

The authors present a comparative study between prostate cancer patients of European and African ancestry, using expression data generated by the commercial Decipher test. The subject of health disparity is important and any additional information on this topic might be relevant.

Although the authors performed a number of analyses, these results are descriptive and not very conclusive. Some additional datasets might be tested for comparison, such as TCGA.

We thank the reviewer for this point. All these prospective samples were from the clinical routine use of the Decipher test. Thus, we have very small number of events (BCR) upon follow-up. Only gene expression data is available for these patients, thus we are limited with the analysis that we can perform.

With regard to comparing our results with TCGA-prostate, We have now conducted differential expression analysis between African American and Caucasians in the TCGA-prostate cohort and found moderate overlap between upregulated and downregulated genes in both cohorts, but the top genes in our cohort were also significantly different in TCGA. We believe our cohort is more powerful to characterize the transcriptomics of prostatic tumor from African Americans as we have over 500 patients compared to ~40 in TCGA.

The validation set is not described at all (numbers, clinical and demographic characteristics, frozen/FFPE samples, RNA quality, experimental details, etc)

We now included more details on the validation cohort. Thank you for your comment.

Methods should present more information, including on all the metrics used in the analyses. It should not take going through references to understand different scores and how they were generated, such as Decipher score, 82 pre-computed gene expression signatures, CARPA-score, pathway activity scores, 20 prognostic signatures, etc. The NCI-60 gene signature analysis is not described in methods at all.

We now added more description on the Drug response score from the NCI-60. We added more reference of our previous work where we used the pathways and signatures.

Supplementary figure 3 should explain error bars and provide p-values

We have now added p values. Data and Error bars are presented as mean \pm SEM and *P<0.05, **P<0.01, ***P<0.001 were considered statistically significant.

Reviewer #3 (Remarks to the Author):

In this manuscript the authors carried out retrospective study by calculating pathway activity scores and multivariable logistic regression analysis using gene expression data derived from microarrays (>46,000 genes), gene expression signatures (82 genes) and Decipher score of 1,152 African-American (AA) and European-American (EA) with prostate cancer who underwent radical prostatectomy. The authors presented results that showed a higher median pre-operative PSA, more triple negative (?), SPINK1 and basal like molecular subtypes among AAs and increased ERG expression among EA. Prostate tumors of AA men were also associated with higher expression FOS and JUN oncogenes, and genes related to inflammation, immune response, hypoxia and reactive oxygen species but lower expression of DNA damage repair and DNA mismatch repair genes. Tumors of EA men, in contrast, were associated with higher levels of expression for fatty acid metabolism, DNA repair and beta catenin signaling. The authors conclude that significant biological differences in prostate cancer appear to be dependent on a man's racial background.

This manuscript proposed that adjuvant treatment after radical prostatectomy can be extrapolated from gene expression signature of prostate tumors. To achieve this the authors described a series of data analyses that associate gene expression profiles, cell signaling pathways, and in vitro drug sensitivity and microarray data from the NCI-60 panel. Overall this idea is original. The manuscript, however, failed to communicate this convincingly because it was neither emphasized in the abstract nor explained clearly in the introduction. Furthermore, missing incomplete citations flagged by placeholders suggest poor editing. The manuscript could be improved if the concerns listed below are addressed.

We thank the reviewer for the summary and we apologize for missing references. We have now reviewed the paper more carefully and made sure there is no missing information. If the reviewer can specify any missing citations we would be happy to add them to the manuscript.

Contrary to the authors assertion (abstract, line 2-3), several studies have characterized racial differences in post prostate cancer at the genomic level through whole exome sequencing and whole genome sequencing of prostate tumors from African American patients. The authors should cite these reports and include them in the introduction or discussion.

We thank the reviewer for highlighting this. We have now added more citations on previous work on racial disparity in prostate cancer especially a new paper on characterizing genomic difference in TCGA-prostate). We also highlighted that all of these studies have small number of African Americans compared to our study that include more than 500 samples from African American patients.

Additional primary references and a link to the website describing the genomic resource information database (GRID) (line 46-53) should be cited.

There is no website for GRID, we have cited our previous work that used GRID signatures.

The authors should provide the number of European American and African American subjects used in the validation assay described on Page 4 (line 72-73), and describe which housekeeping genes used for normalization of PCR assays (Page 6, line 77).
We now included more details on the validation cohort. Data of 14 AAM and 14 EAM have been used. Thank you for your contribution.

Table 1 describing the baseline characteristics should include bladder neck invasion and seminal vesicles invasion (Line91-95).

We realized the discrepancy in terminology between the table and the results. We removed the BNI and SVI terminology to avoid confusion. As per TNM staging, BNI is pT4 and SVI is pT3b disease. Although this data was available to begin with, we have made edits to avoid this confusion. Thank you for your contribution.

Line 96-97/Figure 1: the authors should describe the criteria for selecting the 20 prognostic signatures used in Figure 1 and refer to these reports in the main text.

These are all prostate cancer prognostic signatures from literature that were computed in GRID. We added a citation to our previous work that used these signature. Thank you for your contribution.

It would be helpful if the authors should clarify what does the “triple” in triple negative subtype refer to in Supplementary Figure 1 (line 106-114) and in the abstract.

Triple negative indicates lack of ERG, ETS and SPINK1 expression. We have clarified this in the text. Thank you for your comment.

The authors should include corrected p-values for multiple hypothesis testing to account for false positives or type I error of the multivariate logistic regression analysis in Figure 2 and tabulate the results in the Supplementary data (Page 6, Line 114-124).

We thank the reviewer for this comment. All pvalues reported here are adjusted for multiple testing using the Benjamini-Hochberg procedure. We added a sentence in the methods to highlight that.

It would be helpful if the 82 upregulated genes identified by multivariate logistic regression, together with the p-values correcting for multiple hypothesis testing to account for false positives or type one error were presented in the supplementary data (Line 138-144)

Supplementary table 1 was not mentioned in the text.

We now included a full list of genes upregulated and downregulated in African American in TCGA and our cohort in Supp Table 1. Thank you for your comment.

In addition to reference 27, it would be more relevant if the authors could cite similar epigenetic studies involving human subjects.

The authors should cite relevant references on the factors that potentially contribute to racial health disparities listed in 250 to 255.

The authors should replace the placeholders for references cited at the end of sentences in line 259-262 and 270 with the actual references

We have completed all missing references and missing information. If the reviewer feels that we still have missing citations, we would be more than happy to add them if the reviewer could specify them.

Reviewers' comments:

Reviewer #1 (Remarks to the Author):

There are no additional comments.

Reviewer #2 (Remarks to the Author):

The authors improved their paper by refining some analyses and interpretations, although some methods are still unclear/incomplete and interpretations are not completely justified (at least based on the data and methods shown).

The main conclusion of the paper: Biological characteristics of prostate tumors are substantially different in AAM and EAM. The approaches/findings considered to be novel should be better articulated.

Main points:

1. The external validation set includes only 14 AAM and 14 EAM but provides very strong results presented in Figure S3, with extremely significant p-values (considering the number of samples). At the same time, TCGA-prostate cancer cohort includes 37 AAM and 233 EA, but TCGA set is not used for the analysis presented in Figure S3.

Please show the corresponding results in TCGA. The sample numbers, p-values (the type of statistical analyses, adjustments), and error bars should be explained in each figure. Consider dot-plots instead of bar plots for Figure S3.

2. There is little overlap in the results obtained in this study vs. TCGA. It is also not explained how TCGA analysis was done and what it was adjusted for.

"When conducting a similar 133 differential expression analysis in TCGA-prostate cancer cohort (37 AAM, 233 EA), we found only 35 genes (out of 560 genes upregulated in AAM in TCGA-prostate) in common with the 778 upregulated genes in our cohort, and 295 genes (out of 646 genes with significant lower expression in AAM in TCGA-prostate) were in common with the 3807 genes downregulated in AAM in our cohort".

3. Methods and analyses are still insufficiently described, such as adjustment for covariates in all analyses is mentioned but not explained how it was done:

- Multivariable logistic regression analysis was used to assess the association of race with 82 gene expression signatures. Please show the results for this analysis including values for all covariates as supplementary materials and explain in more detail how it was done.

- For every pathway score, we calculated a regression coefficient after adjusting for clinical and pathological variables. – How the pathways were adjusted for clinical and pathological variables?

4. The original set used FFPE mRNA, the source of mRNA for the validation set is not provided and the quality metrics of the FFPE-extracted RNA are not shown.

5. The top differentially expressed genes are supposed to be shown in Figure 2.A. But Figure 2A annotates the heatmap by publications of unknown significance, but not by genes.

6. What is the purpose and conclusions from Figure 2B analysis on Decipher and Average Genomic Risk analysis?

7. "Differences in tumor-associated intrinsic or extrinsic factors can affect how they respond to drugs. Interestingly, analysis of drug response signatures showed that AAM men have a higher potential response rate to alkylating agent-based chemotherapy, whereas EAM men have a higher response rate to taxane-based chemotherapy".

These predictions are based on NCI60 cell lines and so far are not supported by any clinical observations and these conclusions about "men having a higher response" should be toned down.

8. Will this dataset be deposition into a public database?

Reviewer #3 (Remarks to the Author):

The authors have addressed most of the concerns raised by the reviewers in the revised manuscript. For a few points that were raised by the reviewers that the authors had given their response, the changes, however, was not reflected in the revised manuscript. The authors may missed these minor concerns in their revision.

1. The term "triple negative" used in the abstract and in Supplementary Figure 1 was not clarified in the text to be negative for of ERG, ETS and SPINK1 expression, as claimed in the response by the authors. Perhaps the authors could add this clarification to the text as well as to the legend for Supplementaty figure 1.

2. Under the section "Validation of Differential Immune Response, Immune Response and AR Response Genes" (lines 188-195) and the legend to Supplementary Figure 3, the text in the manuscript and the legend to Supplementary Figure 3 do not match. For example results for CD86, HPGD and PARG are not shown in the figure.

Reviewer #1 (Remarks to the Author):

There are no additional comments.

Reviewer #2 (Remarks to the Author):

The authors improved their paper by refining some analyses and interpretations, although some methods are still unclear/incomplete and interpretations are not completely justified (at least based on the data and methods shown).

The main conclusion of the paper: Biological characteristics of prostate tumors are substantially different in AAM and EAM. The approaches/findings considered to be novel should be better articulated.

Main points:

Concern 1. The external validation set includes only 14 AAM and 14 EAM but provides very strong results presented in Figure S3, with extremely significant p-values (considering the number of samples). At the same time, TCGA-prostate cancer cohort includes 37 AAM and 233 EA, but TCGA set is not used for the analysis presented in Figure S3. Please show the corresponding results in TCGA. The sample numbers, p-values (the type of statistical analyses, adjustments), and error bars should be explained in each figure.

Consider dot-plots instead of bar plots for Figure S3.

Response 1

We appreciate the reviewer's remarks and suggestions with regard to external validation. While we attempted to validate some of our results in the TCGA-Prostate dataset, this effort did not turn out to be meaningful, as they reflect two entirely different cohorts. The TCGA-Prostate dataset cannot be used as a benchmark cohort for racial disparity as only 10% of the cohort (37 patients) is African-American. Our cohort, on the other hand, includes ~500 African-American patients. Additionally, TCGA is based on old tissue analyzed with RNA-seq, while our cohort is of tissue from patients who underwent surgery in the past 2-3 years and analysis was performed on a human exon array platform. Thus, we would not expect to see much overlap between the two cohorts as a result both of sample size and the quality of the tissue.

We have also generated box plots of the genes in the TCGA cohort, which are represented in supplementary figure 2. While some of these are not statistically significant, due to the small sample sizes for AAM, they do trend in the right direction.

The external validation dataset presented as supplementary figure 3, reflects the qPCR analysis of a select number of genes. We have now replaced the figure with dot plots. We would like to clarify that these studies were carried out using tissue from subjects (EAM and AAM) who had prostate surgery and were consented for research. The purpose of this supplemental data is to validate trends in only a select number of genes that were observed in silico/microarray studies as presented in the manuscript. Unlike Decipher, which aggregates a large volume of patient diagnostic data, the availability of research samples is usually very limited. Nonetheless, to

address reviewer concern, we attempted to increase the sample size and validate more genes (DNA mis-match repair and immune related genes) to include in supplementary Figure 3. However, because of the COVID-19 pandemic our laboratories were shuttered, and we were unable to do so.

This information is incorporated here for review

RNA isolated from primary prostate tissues from EAM and AAM subjects (N=11, Gleason Grade Group ≥ 3) was used in qPCR Analysis. Representative genes from DNA repair pathway (MSH2), Inflammation (CCL3, CCL4, IFNB,) and AR response (RLN2, PCGEM1) are shown. Gene expression trends confirms that Immune and AR response genes are upregulated in AAM subjects while DNA mismatch repair gene, MSH2 is downregulated in AAM. Paired t-tests was performed for comparisons (relative target gene expression). Data and Error bars are presented as mean \pm SEM and *P<0.05, **P<0.005 were considered statistically significant.

We are, however, aware of other researchers who are already validating our results in a larger cohort of patients (>17,000) on the same microarray platform as the Decipher GRID. Daniel Spratt et al., DOI:<https://doi.org/10.1016/j.ijrobp.2018.06.104> Abstract Presented at American Society for Radiation Oncology (ASTRO) Annual meeting 2018; manuscript in preparation).

Concern 2. There is little overlap in the results obtained in this study vs. TCGA. It is also not explained how TCGA analysis was done and what it was adjusted for.

"When conducting a similar 133 differential expression analysis in TCGA-prostate cancer cohort (37 AAM, 233 EA), we found only 35 genes (out of 560 genes upregulated in AAM in TCGA-prostate) in common with the 778 upregulated genes in our cohort, and 295 genes (out of 646 genes with significant lower expression in AAM in TCGA-prostate) were in common with the 3807 genes downregulated in AAM in our cohort".

Response 2

Here, again, we appreciate the reviewer's concern regarding analysis of the TCGA cohort. As explained in Response 1, given the vast difference in sample size and the quality of the tissue, we would expect to find little overlap between the cohorts. Our cohort is well annotated with clinical variables, while the TCGA includes very little information on clinical variables. Thus, we used only a Wilcoxon test to assess the differential expression between races. We have added more details about our methods in the manuscript.

We believe, however, that our focus should be on the overlap of the biology between the two cohorts. We cited a recent paper in the section of our manuscript titled "AAM have higher inflammation and immune responses" reporting that immune-related signaling pathways are more active in AAM, which is what we also observed. We also show in Supplemental Figure 2 that DNA repair genes (*MSH2*, *MSHS6*) are downregulated in AAM. Therefore, we believe that despite little overlap in the number of genes, both cohorts show that DNA repair genes are downregulated in AAM and that immune related genes are more active in AAM.

Concern 3. Methods and analyses are still insufficiently described, such as adjustment for covariates in all analyses is mentioned but not explained how it was done:

- Multivariable logistic regression analysis was used to assess the association of race with 82 gene expression signatures. Please show the results for this analysis including values for all covariates as supplementary materials and explain in more detail how it was done.

- For every pathway score, we calculated a regression coefficient after adjusting for clinical and pathological variables. – How the pathways were adjusted for clinical and pathological variables?

Response 3

We apologize for the confusion regarding the gene expression signatures and lack of detail related to analysis. We would like to clarify that we applied logistic regression models to predict race (AAM vs EAM) using pathways (average expression of genes in the pathway) and clinical variables as covariates. We have now added more details about the associations between pathways and race in the manuscript and have added results from UVA and MVA in Supp Table 2.

Concern 4. The original set used FFPE mRNA, the source of mRNA for the validation set is not provided and the quality metrics of the FFPE-extracted RNA are not shown.

Response 4

Decipher has been extensively validated in 43 studies of 30,491 patients and has been demonstrated to be independently prognostic on multivariable analysis with clinical factors and improve discrimination for metastasis and other oncologic endpoints such as recurrence and cancer-specific death. All of these studies were conducted on FFPE samples. The Decipher assay was optimized for use with small amounts of RNA extracted from archived FFPE samples and is the only clinical-grade transcriptome assay for prostate cancer. The Decipher assay is run at the Decipher Biosciences laboratory, located in San Diego, CA, which is certified under the Clinical Laboratory Improvement Amendment (CLIA; 05D2055897), accredited by the College of American Pathologists (CAP, 8859006), and licensed by the New York State Department of Health (NYSDOH, 9018) to run the Decipher FFPE assay.

For validation dataset, RNA was extracted using the Pure-link RNA extraction kit (Thermofisher Scientific) which uses a column extraction method for high quality DNA-free RNA from tissues. Briefly, tissues were homogenized in lysis buffer to ensure RNase-free lysis and purified through a minispin column. Following multiple washes, total RNA was collected into a final volume of 30 ul. Total RNA was measured using nano-drop and 260/230 and 260/280 ratios were considered as measure of quality. We have now added these details to the manuscript.

Concern 5. The top differentially expressed genes are supposed to be shown in Figure 2.A. But Figure 2A annotates the heatmap by publications of unknown significance, but not by genes.

Response 5

We apologize for this confusion which was caused by an inadvertent mislabeling of figures. Figure 2A is now correctly labeled and demonstrates differentially expressed genes.

Concern 6. What is the purpose and conclusions from Figure 2B analysis on Decipher and Average Genomic Risk analysis?

Response 6

We thank the reviewer for this question. The purpose of Fig. 2B is to show that AAM are at higher risk of developing metastasis than EAM only in low Gleason groups based on the genomic signature (Decipher) and that they are at even lower risk (based on average genomic risk) than EAM in high-grade tumors. This is an interesting result as most studies show that AAM are at higher risk of metastasis and death from prostate cancer. The purpose of this figure is to show that genomics may provide a different perspective on why AAM are at higher risk of death from prostate cancer.

Concern 7. "Differences in tumor-associated intrinsic or extrinsic factors can affect how they respond to drugs. Interestingly, analysis of drug response signatures showed that AAM men have a higher potential response rate to alkylating agent-based chemotherapy, whereas EAM men have a higher response rate to taxane-based chemotherapy".

These predictions are based on NCI60 cell lines and so far are not supported by any clinical observations and these conclusions about "men having a higher response" should be toned down.

Response 7

As requested, we have toned down our conclusions regarding predicted response to chemotherapy drugs in the Discussion section of the manuscript.

Concern 8. Will this dataset be deposition into a public database?

Response 8

This data will be available from the PI of the study upon request. It is also freely available through the Decipher GRID program upon request from Decipher Biosciences. The gene expression data presented in our manuscript are available on the Decipher GRID registry, and researchers have started using this cohort to further understand the biology of tumors in African-American prostate cancer patients (Daniel Spratt et al., DOI: <https://doi.org/10.1016/j.ijrobp.2018.06.104> (Abstract Presented at American Society for Radiation Oncology (ASTRO) Annual meeting 2018; manuscript under preparation.)

Reviewer #3 (Remarks to the Author):

The authors have addressed most of the concerns raised by the reviewers in the revised

manuscript. For a few points that were raised by the reviewers that the authors had given their response, the changes, however, was not reflected in the revised manuscript. The authors may missed these minor concerns in their revision.

Concern 1. The term "triple negative" used in the abstract and in Supplementary Figure 1 was not clarified in the text to be negative for of ERG, ETS and SPINK1 expression, as claimed in the response by the authors. Perhaps the authors could add this clarification to the text as well as to the legend for Supplementary figure 1.

Response 1

We appreciate this positive overall assessment of our paper and the suggestions provided to help further improve the manuscript. Patients that are ERG-, SPINK1- and ETS- are referred to as Triple negative. As requested, we have now added the definition of TripleNeg in the legend of Figure S1.

Concern 2. Under the section "Validation of Differential Immune Response, Immune Response and AR Response Genes" (lines 188-195) and the legend to Supplementary Figure 3, the text in the manuscript and the legend to Supplementary Figure 3 do not match. For example results for CD86, HPGD and PARG are not shown in the figure.

Response 2

We apologize for this oversight. We have now ensured that gene names in the legend and the figure match and that the list is complete.

Reviewers' comments:

Reviewer #1 (Remarks to the Author):

There are no additional comments.

Reviewer #2 (Remarks to the Author):

Thank you for providing the responses and some clarifications. I provide additional comments in corresponding sections.

Concern 1. In the previous version, the external validation set was presented as 14 AAM and 14 EAM. In the current version, these numbers are 11 and 11. Still, the results are shown as significant for one gene and borderline significant for 3 more genes tested. However, these p-values are based on paired T-tests. These are two different sample sets, which are not paired and this test is not appropriate. This should be at least a two-sided non-paired T-test or a corresponding non-parametric test.

11 AAM and 11 AEM samples for validation are considered OK while 37 AAM vs. 233 EAM in TCGA are considered too few? The true results should be validated regardless of the platform (RNA-seq) of exome arrays. The source of RNA for validation samples is still not provided - was it fresh-frozen or FFPE tissue?

Concern 2. It is known and expected that tumors from AAMs are more inflamed. The results of this paper should be independent of what is expected and have to present enough evidence to defend and extend this conclusion.

Table S1 presents the differences between expression in AAM and EAM. FDR-adjusted p-values in TCGA should also be presented. More importantly, all the effects in TCGA are in the direction opposite to observed in this study. TCGA has enough covariates that can be used trying to explain this complete lack of validation. But this is not demonstrated our commented on.

Concern 3. Logistic regression was used to predict race based on the expression of all the genes in pathways. Having race as a dependent (response) variable doesn't make any sense. Race does not respond to or is modified by any of the variables tested. A multivariable linear regression expression analysis with race and any other factors as covariates would be a more relevant approach, which should identify expression of genes/pathways significantly affected by race. Alternatively, a stratified analysis in AAM and AEM could be used.

How exactly was the average expression of all the genes in pathways analyzed? Were the same genes included in different pathways, etc?

Concern 5. The heatmap with publications mentioned before as part of Figure 2A, was reported to be replaced by an expected heatmap with genes. However, the same heatmap with publications is now used as Figure 1A.

Concern 7. Response to treatments. Still, the language used indicates "EAM men have a higher response to xx chemotherapy". This is based on scores calculated based on NCI-60 cell lines and no treatment outcomes were analyzed here.

Concern 8. The response on data sharing provided in the rebuttal is not mentioned in the paper.

Please comment on:

Ln 196. "CRYBB2 and IL20RA are expressed in the eye lens and skin respectively, supporting their potential role in discriminating between AAM and EAM tumors".

p. 321. "AAM patients have increased susceptibility to DNA-damaging radiation therapy".

Point-by-point response to all of the concerns raised by the reviewer 2.

Concern 1. In the previous version, the external validation set was presented as 14 AAM and 14 EAM. In the current version, these numbers are 11 and 11. Still, the results are shown as significant for one gene and borderline significant for 3 more genes tested. However, these p-values are based on paired T-tests. These are two different sample sets, which are not paired, and this test is not appropriate. This should be at least a two-sided non-paired T-test or a corresponding non-parametric test.

Response 1. Reviewer 2 has misunderstood the reason for the selection of 11 cases vs 14. In the previous submission, reviewer 2 had concerns that the number of samples was small to see significant changes. At that time, we had 5 samples with grade group 1, 5 samples with grade group 2, and 4 with higher grade. For this revision, we actually increased the number to 11 and included all patients in grade group 3 and above. In summary, for the previous submission, we had 9 cases for Gleason Group 3 and above and in this revision, we had 11. Non-paired T-test has been used for analysis and results are presented in Supplemental Figure 3.

Concern 2. 11 AAM and 11 AEM samples for validation are considered OK while 37 AAM vs. 233 EAM in TCGA are considered too few? The true results should be validated regardless of the platform (RNA-seq) of exome arrays. The source of RNA for validation samples is still not provided - was it fresh-frozen or FFPE tissue?

Response 2. We think the reviewer has clearly overlooked this explanation in our appeal/rebuttal letter. I am including it here again for your review.

Reviewer 2's remarks mainly concern the validation of our results in the TCGA-prostate dataset in which only 37 (out of 333) are self-identified African-American patients. Our cohort has more than 500 African American patients (~50% of the cohort) with genome-wide expression profiles and, to the best of our knowledge, represents the largest cohort of African American gene expression related to prostate cancer in a single platform. We suggest that TCGA should not be considered the benchmark for racial disparities in prostate cancer, but that our cohort should instead perhaps be considered the benchmark.

Regarding the reviewer's question about the source for RNA samples used for validation, we have provided the details in the manuscript on Page # 6. The text is included here for

For external validation of nominated targets in other platform qPCR, RNA isolated from fresh primary prostate cancer tissues obtained from EAM and AAM (n=11) who consented to PPHS/IRB study (Mount Sinai # GCO 14-0318) were used. Total RNA was prepared using Purelink RNA mini kit (Invitrogen) and reverse transcribed using the iscript Advanced cDNA synthesis reagent's following the manufacturer's instructions (BioRad Laboratories, USA). Tissues were homogenized in lysis buffer to ensure RNase-free lysis and purified through a mini spin column. Following multiple washes, total RNA was collected into a final volume of 30 ul. Total RNA was measured using nano-drop and 260/230 and 260/280 ratios were considered as measure of quality. Quantitative real-time PCR was performed using SSO-Advanced Universal SYBR Green supermix (BioRad Laboratories, USA) and was analyzed on the CFX384 Touch

real-time PCR system (BioRad Laboratories). Transcript levels were measured using the Delta-Ct method after normalization to seven housekeeping genes (ACTB, HMBS, RPL38, TBP, GAPDH, HPRT, PSMC1) (1, 2).

We also describe the quality metrics for RNA from fresh tissues is given below and seen on page # 6

Validation of nominated targets in EAM and AAM subjects

To further validate the differential expression of nominated targets in AAM vs EAM we used quantitative PCR to analyze transcript profiles of representative genes that belong to DNA repair pathway (MSH2), Inflammation (CCL3, CCL4, IFNB), and AR response (RLN2, PCGEM1). RNA was extracted using a Pure-link RNA extraction kit (Thermofisher Scientific) which uses a column extraction method for high-quality DNA free-RNA from tissues. Briefly, tissues were homogenized in lysis buffer to ensure RNase free lysis and purified through mini spin column. Following multiple washes, total RNA was collected into a final volume of 30 ul. Total RNA was measured using nano-drop and 260/230 and 260/280 ratios were considered a measure of quality. qPCR findings confirmed that genes associated with inflammation and AR activity are upregulated in AAM and DNA mismatch repair gene, MSH2 is downregulated in AAM (Supplemental Figure 3).

Concern 3. It is known and expected that tumors from AAMs are more inflamed. The results of this paper should be independent of what is expected and have to present enough evidence to defend and extend this conclusion.

Response 3: First of all, this paper focuses not only on the inflammation of the tumor across race. We conducted differential expression across all genes and curated pathways and independently found that inflammation-associated genes are over-expressed in AAMs. Second, we do not understand why it is “known and expected” that tumors from AAM are more inflamed than EAM? Most of the work to date has speculated a role for acute or chronic inflammation, but none of the work has provided direct evidence or addressed the role of inflammation associated genes/pathways in a cohort of ~ 1000 patients with more than 500 AAM. We believe that the conclusions derived from this significant cohort is enough evidence.

Concern 4: Table S1 presents the differences between expression in AAM and EAM. FDR-adjusted p-values in TCGA should also be presented. More importantly, all the effects in TCGA are in the direction opposite to observed in this study. TCGA has enough covariates that can be used trying to explain this complete lack of validation. But this is not demonstrated our commented on.

Response 4: We have added FDR adjusted p-value to Table S1. It is not true that “all the effects” in TCGA are in the opposite direction to what we observe in our cohort.” We already showed in Figure 2 that the most differentially expressed genes in our cohort are also differentially expressed and in the same direction in TCGA. As we mentioned earlier, the TCGA cohort should not be used as a benchmark for racial disparity studies as AAM are underrepresented in that study.

We are already working with other groups to assess the clinical implications of the molecular differences between AAM and EAM. In three independent cohort (total n >30,000), we observed similar biological differences across race, and the same biological pathways are differentially expressed. Unfortunately, this is part of another clinical story that is not part of this work.

Concern 5. Logistic regression was used to predict race based on the expression of all the genes in pathways. Having race as a dependent (response) variable doesn't make any sense. Race does not respond to or is modified by any of the variables tested. A multivariable linear regression expression analysis with race and any other factors as covariates would be a more relevant approach, which should identify expression of genes/pathways significantly affected by race. Alternatively, a stratified analysis in AAM and AEM could be used.

Response 5: The reviewer is confused about the rationale of using logistic regression. We are not trying to build a logistic regression model to collectively identify pathways in a multivariable setting that best predicts race. We used logistic regression with race as a response variable to assess the significant association between pathway activity score (as an average of gene expression within the pathway) and not as a set of genes. We did this analysis for each pathway individually to find the statistical associations between the pathway activity score and race (binary response). We believe this is a straightforward statistical problem where we have a binary variable (race) and a continuous variable (pathway activity). We use logistic regression with race as a response variable to find the statistical association between the two variables.

Using the multivariable linear regression where pathway activity is the response variable suggested by the reviewer, it is not appropriate for finding how each pathway activity is different across the race.

Concern 6. How exactly was the average expression of all the genes in pathways analyzed? Were the same genes included in different pathways, etc?

Response 6: Expression of genes within a single pathway was first standardized, then the mean of expression of all genes in individual pathways was used to represent the pathway activity. Some genes may play a role in multiple pathways. All pathways were obtained from the Molecular Signatures Database (MSIGDB) (<https://www.gsea-msigdb.org/gsea/msigdb/index.jsp>).

Concern 7. The heatmap with publications mentioned before as part of Figure 2A, was reported to be replaced by an expected heatmap with genes. However, the same heatmap with publications is now used as Figure 1A.

Response 7: We believe the reviewer is confused here. The heatmap of existing prognostic signatures is now shown in Figure 1A and bar plot of differentially expressed genes is shown in Figure 2A. We didn't use heatmap here as bar plot better shows order of genes of their differential expression.

Concern 8. Response to treatments. Still, the language used indicates "EAM men have a higher response to xx chemotherapy". This is based on scores calculated based on NCI-60 cell lines and

no treatment outcomes were analyzed here.

Response 8. The reviewer raised this concern before, and we changed the language in the paper. This is what we wrote in the paper -results “These signatures predicted AAM men will have higher response score to DNA damage and alkylating agent-based chemotherapy and EAM men to potentially respond better to anti-microtubule-based chemotherapy”. But it seems we missed that sentence in the discussion. We apologize for that.

Concern 9. The response on data sharing provided in the rebuttal is not mentioned in the paper.

Response 9: We apologize for the overlook. We have included this in the revised manuscript and also presented here.

This data will be available from the PI of the study upon request. It is also freely available through the Decipher GRID program upon request from Decipher Biosciences. The gene expression data presented in our manuscript are available on the Decipher GRID registry.

Researchers have started using this cohort to further understand the biology of tumors in African-American prostate cancer patients (Daniel Spratt et al., DOI:<https://doi.org/10.1016/j.ijrobp.2018.06.104>

(Abstract Presented at American Society for Radiation Oncology (ASTRO) Annual meeting 2018; manuscript under preparation.)

Please comment on:

Ln 196. “CRYBB2 and IL20RA are expressed in the eye lens and skin respectively, supporting their potential role in discriminating between AAM and EAM tumors”.

Response: As noted on page # 9

CRYBB2 is one of the most up-regulated genes in the AAM group in both the TCGA-prostate cohort and our cohort. Other genes, including *GSTM3*, *SNX31* and *TENM1*, were among the top genes upregulated in AAM in both cohorts (**Figure 2.B**). *CRYBB2* and *PSPH* had been previously reported to be overexpressed in breast cancer samples (3) and colorectal cancer samples from AAM patients (4). This suggests that these genes are overexpressed in AAM race regardless of cancer type or tissue of origin.

p. 321. “AAM patients have increased susceptibility to DNA-damaging radiation therapy”.

Response: Our work here shows that DNA repair are less active in AAM, thus suggesting that AAM could respond better to Radiation therapy. Indeed, recent work by Spratt et al, <https://doi.org/10.1016/j.ijrobp.2018.06.104>, showed that AAM respond better to Radiation therapy. Thus, our work here proves a biological explanation for this clinical observation.

References

1. Nair SS, Li DQ, Kumar R. A core chromatin remodeling factor instructs global chromatin signaling through multivalent reading of nucleosome codes. *Mol Cell*. 2013;49(4):704-18.
2. Chakravarty D, Sboner A, Nair SS, Giannopoulou E, Li R, Hennig S, et al. The oestrogen receptor alpha-regulated lncRNA NEAT1 is a critical modulator of prostate cancer. *Nat Commun*. 2014;5:5383.
3. Field LA, Love B, Deyarmin B, Hooke JA, Shriver CD, Ellsworth RE. Identification of differentially expressed genes in breast tumors from African American compared with Caucasian women. *Cancer*. 2012;118(5):1334-44.
4. Jovov B, Araujo-Perez F, Sigel CS, Stratford JK, McCoy AN, Yeh JJ, et al. Differential gene expression between African American and European American colorectal cancer patients. *PLoS One*. 2012;7(1):e30168.

Reviewers' comments:

Reviewer #4 (Remarks to the Author):

Thank you for submitting an excellent paper.

I suppose the difference between EAM and AAM should be of great interest for readers.

Major comments

1. RNA differential expression study (DES) between EAM and AAM PCa

When performing RNA DES between 2 groups, two thresholds, FDR and fold change, are usually used to choose genes with significant altered expression. Then, volcano plot should be shown as a figure.

e.g. <https://training.galaxyproject.org/training-material/topics/transcriptomics/tutorials/rna-seq-viz-with-volcanoplot/tutorial.html>

(1) Page.8 line.190 Using and adjusted P-value of $1e-5$ as a cut-off,--

Why did you use just a p-value threshold for DES. Why you did not use a fold change threshold? Please describe the rationale in the method.

(2) Page.8 line.190 Using and adjusted P-value of $1e-5$ as a cut-off,--

Considering the nature of multiple testing correction for RNAseq, the threshold "p-value of $1e-5$ " looks lax (equal to bonferroni correction for 5000 genes). This p-value threshold is derived from FDR calculation? If so, please show us the FDR threshold. $FDR < 0.05$ or $FDR < 0.1$?

Basically, p-value and FDR should be presented separately like the paper

<https://www.nature.com/articles/nature13772>.

FDR is a FDR. Your "FDR-adjusted p-value" is very confusing.

(3) "Mean difference" Metric in figures and supplementary tables

As I mentioned, "fold change" is often used in RNA DES.

In normalized parameters, how should we understand the meaning of "Mean difference"? I understand the direction is important, but how about the magnitude of the value?

2. Page.9, Line .210 Pathway activity was summarized as the mean expression of genes in the pathway.

I have never seen this method. I am wondering whether "summarized as the mean" is on the right track or not. Please explain the rationale of the method and refer to some paper this method used.

3. Reviewer2. Multivariate logistic regression analysis for gene expression and pathways.

For readers' better understanding, could you please show us the formula of the logistic regression in the method section and which p-value was evaluated in each analysis?

e.g. EAM or AAM (Outcome variable) $\sim b_0 + b_1 * \text{pathway} + b_2 * \text{covariates (Cancer stage,, etc)} + b_3 * \dots$. Then p-value for b_1 was evaluated in the analysis.

4. Overall, methods are difficult to understand. Please describe each method in more detail, or refer to previous papers the method was used.

Minor comments

1. Page.9, Line 193 "(37 AAM, 233 EA)" should be "(37 AAM, 233 EAM)".

2. Page.13, Line 308 "PCa beteen EAM and EAM" should be "PCa between EAM and AAM".

Response to Reviewers

Reviewer #4

Major comments

1. RNA differential expression study (DES) between EAM and AAM PCa

When performing RNA DES between 2 groups, two thresholds, FDR and fold change, are usually used to choose genes with significant altered expression. Then, volcano plot should be shown as a figure.

e.g. <https://training.galaxyproject.org/training-material/topics/transcriptomics/tutorials/rna-seq-viz-with-volcanoplot/tutorial.html>

(1) Page.8 line.190 Using and adjusted P-value of $1e-5$ as a cut-off,--

Why did you use just a p-value threshold for DES. Why you did not use a fold change threshold? Please describe the rationale in the method.

(2) Page.8 line.190 Using and adjusted P-value of $1e-5$ as a cut-off,--

Considering the nature of multiple testing correction for RNAseq, the threshold "p-value of $1e-5$ " looks lax (equal to bonferroni correction for 5000 genes). This p-value threshold is derived from FDR calculation? If so, please show us the FDR threshold. $FDR < 0.05$ or $FDR < 0.1$?

Basically, p-value and FDR should be presented separately like the paper <https://www.nature.com/articles/nature13772>.

FDR is a FDR. Your "FDR-adjusted p-value" is very confusing.

(3) "Mean difference" Metric in figures and supplementary tables

As I mentioned, "fold change" is often used in RNA DES.

In normalized parameters, how should we understand the meaning of "Mean difference"? I understand the direction is important, but how about the magnitude of the value?

Response:

We thank the reviewer for the suggestion. We would like first to bring the reviewers attention to the expression data we are using in this paper. Expression data here is obtained from Affymerix Human Exon ST 1.0 array that is normalized using an algorithm called Single Channel Array Normalization (SCAN). Thus, the distribution of the expression levels is different that RNAseq and thus we employed different methods.

We now have generated a volcano plot and added it to supplementary figures.

We have also replaced adjusted p value with FDR in tables and text as the adjusted p value was generated from FDR calculations.

For FDR threshold, we used very stringent value ($1e-5$) as sample size was large in this analysis (596 AAM vs 556 EAM) and thus any small difference will lead to significant p-value. If we use FDR (0.01), we end up with ~12,000 genes as differentially expressed. We provided all genes with $FDR < 1e-5$ in Table 1 to allow readers to filter genes as they see appropriate.

For mean difference metric, we found mean difference is more appropriate than fold change as this data is not RNA-seq and thus we tweaked some methods to better fit it. Overall mean of values in this data is 0.5 and thus fold change ratio leads to very high values that has no biological interpretation. Mean difference here just indicates directionality and reader should rely on FDR for significance.

2. Page.9, Line .210 Pathway activity was summarized as the mean expression of genes in the pathway.

I have never seen this method. I am wondering whether "summarized as the mean" is on the right track or not. Please explain the rationale of the method and refer to some paper this method used.

Response:

For every pathway with multiple genes, we first scaled the expression of the genes around the median and then took the mean. We have used this method in couple of papers before [Kishan et al, European Urology, 2020; Labbe et al, Nature Communications, 2020; Spratt et al, Clinical Cancer Research, 2020]. We now have cited some of these in the methods section.

3. Reviewer2. Multivariate logistic regression analysis for gene expression and pathways.

For readers' better understanding, could you please show us the formula of the logistic regression in the method section and which p-value was evaluated in each analysis?

e.g. EAM or AAM (Outcome variable) ~ $b_0 + b_1 * \text{pathway} + b_2 * \text{covariates (Cancer stage, etc)} + b_3 * \dots$. Then p-value for b_1 was evaluated in the analysis.

Response:

We thank the reviewer for this suggestion. We now included the logistic regression formula in the methods section.

Response (1: AAM, 0: EAM) ~ $b_0 + b_1 * \text{pathway} + b_2 * \text{Gleason} + b_3 * \text{EPE} + b_4 * \text{SVI} + b_5 * \text{LNI}$

4. Overall, methods are difficult to understand. Please describe each method in more detail, or refer to previous papers the method was used.

Response:

We have added more details to some methods sections and cited some of other papers where they used similar methods.

Minor comments

1. Page.9, Line 193 "(37 AAM, 233 EA)" should be "(37 AAM, 233 EAM)".

2. Page.13, Line 308 "PCa beteen EAM and EAM" should be "PCa between EAM and AAM".

Response:

We apologize for the typos. Both cases are corrected and we made sure to use AAM and EAM across the whole paper.

REVIEWERS' COMMENTS:

Reviewer #4 (Remarks to the Author):

Thank you for your revised manuscript.
Overall, I'm satisfied. But I have just one thing to make sure.

Major Concern:

For FDR calculation, what method did you use? Benjamini-Hochberg, adaptive Benjamini-Hochberg, Benjamini-Yekutieli or Storey method?

Also, how did you calculate FDR (a.k.a q-value) in concrete?

To calculate FDR, p-values and the number of test are required as shown in the below link (BH method).

<https://stats.stackexchange.com/questions/238458/whats-the-formula-for-the-benjamini-hochberg-adjusted-p-value>

What is the number of test (represented as the variable "m" in the above link) in your analysis?

Minor:

Page.7. Formally, beta in the formula should be presented as not b but β . (In my previous response, I skimmed the expression and wrote "b" for beta. But formally, we should use β)

So please change the formula

"(1: AAM, 0: EAM) $\sim b_0 + b_1 \cdot \text{pathway} + b_2 \cdot \text{Gleason} + b_3 \cdot \text{EPE} + b_4 \cdot \text{SVI} + b_5 \cdot \text{LNI}$ "

to

" $\log(p/1-p)$ (p=1:AAM, 0:EAM) = $\beta_0 + \beta_1 \times \text{pathway} + \beta_2 \times \text{Gleason} + \beta_3 \times \text{EPE} + \beta_4 \times \text{SVI} + \beta_5 \times \text{LNI} + \epsilon$ "

where $\log(p/1-p)$ is a logit function and ϵ is an error for logistic regression formula.

[Editor note: In follow-up correspondence, the reviewer checked the preliminary revised version and agreed that all concerns had been addressed]